# Trusted Blockchain-Driven IoT Security Consensus Mechanism

**Chuansheng Wang [1], Xuecheng Tan [1,\*], Cuiyou Yao [1], Feng Gu [2], Fulei Shi [1] and Haiqing Cao [1]**

1   School of Management and Engineering, Capital University of Economics and Business,
    Beijing 100070, China; wangcs@cueb.edu.cn (C.W.); ycy@cueb.edu.cn (C.Y.); shifulei@cueb.edu.cn (F.S.);
    caohaiqing@cueb.edu.cn (H.C.)
2   Department of Computer Science, College of Staten Island, The City University of New York,
    New York, NY 10314, USA; feng.gu@csi.cuny.edu
\*   Correspondence: stefan@cueb.edu.cn

**Abstract:** Single point of failure and node attack tend to cause instability in the centralized Internet of Things (IoT). Combined with blockchain technology, the deficiency of traditional IoT architecture can be effectively alleviated. However, the existing blockchain consensus mechanism still has the problems of forks and wasting of computing power. Therefore, this paper proposes a new framework based on a two-stage credit calculation to handle these problems. Notably, the nodes are selected through the model, and these nodes will compete on the chain according to the behavior of participating in the creation of the block. Then, a comparative simulation with the existing consensus mechanism proof of work (PoW) is presented. The results show that the proposed framework can quickly eliminate malicious nodes, maintain the overall security of the blockchain and reduce consensus delay.

**Keywords:** blockchain; IoT; rrust model; PoT

## 1. Introduction

IoT connects physical devices to the internet through infrared sensors, radio frequency identification, and global positioning systems to achieve data exchange and communication, thereby completely breaking data flow and data islands [1]. At present, most of the information interacted by IoT devices is stored through third-party intermediaries, but the cost of centralized storage is high and there is a risk of data leakage [2]. First, data are not managed directly by itself, but stored on centralized platforms, which require high costs for safe operation [3]. Second, these centralized control service providers, governments, and manufacturers can collect and investigate user information, thereby increasing the risk of data leakage. It is worthy to note that smart devices and user data can only be managed by central nodes and are vulnerable to attacks. Once the central node is destroyed, the entire system is paralyzed and a large amount of personal data will be attacked [4].

In recent years, blockchain as a decentralized technology has gained considerable interest. Blockchain has the advantages of decentralization, encryption algorithm, immutable information, consensus mechanism, etc. [5–7]. Once the information on the chain is verified by the consensus node, it will be stored forever unless more than 51% of the nodes on the chain can be controlled at the same time. Therefore, by combining the blockchain consensus mechanism with the IoT, there is an opportunity to solve the security problems caused by the abovementioned centralized data storage [8–10].

The existing researches focus on traditional consensus mechanism, such as PoW [11], proof of stake (PoS) [12] and delegated proof of stake (DPoS) [13]. Then, some researchers try to build blockchain consensus mechanisms by combining different consensus mechanisms, such as bitchin-NG [14], proof of elapsed time (Poet) [15] and proof of voting (PoA) [16]. The blockchain consensus mechanism effectively solves the problem of node trust in distributed systems. Thus, the security problems of network collapse caused by a

single point of failure and the data leakage caused by attacks on the central node can be overcome [17]. Additionally, the blockchain is a distributed ledger maintained by multiple points, eliminates the need to maintain an intermediary and saves costs. However, most of the existing consensus mechanisms use computing power on the chain, which leads to problems with forks and consensus delay. Forks lead to instability and insecurity of the overall blockchain. Consensus delay does not apply to the wider IoT application areas. Therefore, the aim of finding a balance between efficiency and safety has faced challenges [18].

To address the above-mentioned challenges, this paper conducts a blockchain-based IoT system, which includes the proof of trust (PoT) consensus mechanism based on credit model. Firstly, we select candidate nodes with the capable of generating blocks through the credit model to reduce the security risks caused by collusion. Secondly, the candidate nodes compete upchain based on the behavior of participating in block-creation, and adaptively adjust the difficulty of mining to reduce the inefficiency caused by the consensus delay. Finally, experiments are conducted to demonstrate the superiority of the blockchain-based IoT system.

Our main contributions of this paper are described as follows:

(1) Proposing a two-stage credit evaluation model to calculate the credit value of each node. In the first stage, the initial trust value is obtained through historical communication between nodes. In the second stage, the recommendation of other nodes is considered, and the neighborhood is established according to the communication radius of the node. The initial trust value is verified by the recommended information, and the trust value of the node is dynamically updated.

(2) Introducing a PoT consensus mechanism based on credit model, in which a reward and punishment mechanism is established. According to the node behavior, honest behavior is rewarded and malicious behavior is punished. In particular, the mining difficulty is adjusted according to the credit value obtained by the node, instead of computing power mining.

(3) Building the system framework of the IoT based on PoT consensus mechanism, and then test the existing consensus mechanism PoW with our proposed PoT by simulation. The experimental results show that the proposed PoT can quickly eliminate malicious nodes and effectively reduce consensus delay.

The remainder of this paper is organized as follows. Section 2 introduces the background of blockchain technology combined with the Internet of Things. Section 3 presents blockchain IoT framework, including PoT consensus mechanism and credit model. Section 4 simulates the proposed PoT consensus mechanism and demonstrate the idea through comparative experiments. Finally, Section 5 summarizes this paper.

## 2. Literature Review

Blockchain is a distributed ledger maintained by multiple points, which is supported by encryption algorithms and consensus mechanisms. Decentralized blockchain enable parties which do not fully trust each other to maintain consensus about records, storage and update of a set of transaction. The consensus mechanism plays a decisive role in the establishment of blockchain trust. It mainly refers to the process by which all nodes in the network reach agreement on a proposal through a specific rule.

Barahanpure et al. [19] proposed a proof of stack algorithm, assuming that the chance of a node obtaining mining rights is proportional to the trust that the node obtains in the entire blockchain community, which makes up for the waste of computing power. Wang et al. [20] proposed a consensus mechanism POC based on reputation accumulation model, in which the trust value of nodes can be calculated to help select trusted block producers. Li et al. [21] established a proof-of-voting consensus algorithm (POV), wherein nodes can vote according to the historical transaction records of the target node in the network, and they can be uploaded to the chain through verification in other nodes. However, the algorithm cannot guarantee efficiency, and there are situations where nodes collude to do evil. Li et al. [22] proposed a scalable and practical Byzantine Fault Tolerance

(EPBFT) algorithm. However, there are cases where too many malicious nodes lead to consensus. In the distributed network of IoT devices, Di Pietro et al. [23] proposed Trust chain, which replaces PoW by establishing a trust mechanism to enable nodes on the chain to provide security services and be more able to resist attacks. Reyan et al. [24] applied the improved blockchain consensus mechanism to the IoT to realize the interoperability of the IoT system and improve the security, reliability and scalability. Lu et al. [25] developed a blockchain-based traceability system product to provide tracking services for suppliers and retailers, and ensure the reliability of IoT data through the non-tamperable modification of the blockchain. Wan et al. [26] proposed an automated production platform based on distributed blockchain, which provides better security and privacy protection than the traditional centralized architecture. The above documents proposed the application scenarios and achievable solutions for applying blockchain to the IoT. However, there are still some problems about the balance between efficiency and security in the current application of blockchain. The following documentation described the solution and provided some ideas.

Wang et al. [27] proposed a consensus protocol based on the credit model. First, the consensus protocol draws on the idea of personal credit risk assessment, and designs a node credit model based on BP neural network. Secondly, a sharing rotation model is constructed, which can split the search space to generate new blocks according to the creditworthiness of the nodes, and at the same time analyze the possible attacks faced by the protocol, and fix the loopholes in the protocol. Qiu et al. [28] presented the security and trustful issues related to blockchain from the deep learning perspective. Meng et al. [29] established a Bayesian model and proposed the assumption that there are 1/2 error nodes in the network. According to the model, the error nodes are gradually eliminated to improve the reliability of trust management, and the model was verified in two hospitals. This scheme can well realize the deletion of abnormal points, but the efficiency of the blockchain is not discussed. Fortino et al. [30] built a reputation model by establishing an agent community, which realized that IoT devices can meet more honest partners when they migrate in the environment, and avoid the security problem of single point of failure. Schooler et al. [31] established a reputation security framework, in which the recent performance of the device can be inferred to confirm the trust degree based on the historical behavior of the device, and the IoT device associated with it can be found and assigned to the same set. Alshehri et al. [32] established a trustworthy environment by considering the large-scale nature of IoT nodes, and considered the negative evaluation of trust in the environment to effectively identify malicious node attacks.

In summary, the existing research recognized the advantages that blockchain potentially addressing single points of failure and lack of trust in IoT. However, initial implementations are limited in two ways. First, they still have collusion evil, which reduces the credibility of honest nodes and put the security of the entire blockchain at risk. Second, they only solve the problem of lack of trust without consideration for efficiency, leaving many power-constrained IoT devices unable to deliver information in a timely manner, which causing consensus delays and leading to instability of the entire blockchain. Therefore, in order to effectively balance safety and efficiency, combined with the overview of existing articles, this paper presents an improved PoT consensus mechanism.

## 3. The Proposed Trust Model

In this section, a decentralized IoT framework based on the trust consensus mechanism (PoT) is presented in Figure 1. In the PoT-IoT framework, each entity represents a node in the blockchain. According to the performance of the node, it is divided into two types of nodes, namely ordinary nodes and special nodes. Ordinary nodes (external sensors) can enter and exit the blockchain network to obtain information at will, but they do not participate in block creation activities. Special nodes (servers and gateways) have the characteristics of long-term online, fast processing speed, and high security, which are used for block generation and blockchain community maintenance. Through the communication between these nodes, a new IoT framework is built. The detailed design of PoT-IoT system

framework is introduced from two parts, including trust model construction and consensus mechanism establishment.

**Figure 1.** IoT-PoT framework.

A two-stage credit value calculation model is established. In the first stage, the initial trust value is obtained through the historical communication of nodes. In the second stage, the initial trust value is verified according to the recommendation of other nodes, then, the comprehensive trust value of nodes is calculated. Therefore, the connection mode of nodes needs to be redefined.

In distributed network, nodes realize the calculation of trust value from one node to another node in self-organization mode. The evaluation formed from the results of node's direct interaction behavior is direct trust, and the evaluation formed from the recommendation of node's common neighbor is indirect trust. The node calculates the initial trust value through historical communication and stores it in the blockchain for broadcasting so that all nodes can obtain the trust situation of the target node. Establishing neighborhoods can prevent nodes from conspiring to increase their trust value to seek benefits on the block. Communication behavior of the same node in the neighborhood approximately obeys the normal distribution. According to the normal distribution, the trust value of nodes in the same neighborhood for the communication behavior of the target node is concentrated, that is, the trust value evaluation of most nodes in the same neighborhood on the communication behavior of the target node will not deviate too much. If a node promotes its trust value through its good performance in the previous consensus process and launches a malicious recommendation attack, the node will be identified as a malicious node in the subsequent communication process. Therefore, the dynamic behavior update of nodes can be achieved through the two-stage trust value calculation.

### 3.1. Parameter Settings and Blockchain-Based Trust Calculation

This paper randomly simulates 100 nodes, among which 20 nodes are selected as consensus nodes based on the trust value. These nodes are responsible for maintaining the overall security and testing the proposed trust framework by changing the proportion of malicious nodes. The setting of the main parameters adopted in these simulations were are given in Table 1.

This article carries out the following steps in the credit model node connection.

(1) 100 IoT devices node interact with each other to select 20 block-producing nodes.

(2) Interactions were arranged in 20 epochs and, in turn, node information is updated in each epoch.

(3) Set up four recommended nodes around the consensus node, and perform initial information verification based on the similarity of the recommendations.

(4) Set initial node parameters $\alpha$ and $\beta$ according to different scenarios, and calculate the initial trust value of the node. Here, we set it as 0.5.

(5) Node interaction has timeliness. In order to be closer to the real environment, we set the time decay parameter.

(6) We assume that the proportion of malicious nodes in the environment ranges from 30% to 60%, the proportion is constantly increasing to test the stability of the mechanism.

**Table 1.** Setting parameters.

| Symbol | Description |
| --- | --- |
| i,j | i represents the target node and j represents the node interacting with target i |
| m,x | m represents the number of recommend nodes, and x represents the number of consensus rounds |
| S | The voting nodes cast an honest vote in the consensus process |
| F | Votes cast maliciously by a voting node during consensus |
| A | The number of honest signatures during the voting cycle |
| B | Number of malicious signatures in the voting cycle |
| H | Time forgetting factor, to avoid the rapid growth of trust, to prevent centralization |
| $\Theta$ | Vicious voting penalty factor |
| $\Lambda$ | Depending on the environment, the degree of dependence on the initial trust value is set |
| H | The weight of historical communication |
| DP | Initial trust value |
| IP | The evaluation value of the nodes in the neighborhood of the target node |
| P | Node trust value |
| $\Delta T$ | Denotes a unit of time |
| K | The number of valid transactions of target node |
| W | Prevent the trust value of a node from growing too fast |

The trust value of nodes follows the Beta ($\alpha$, $\beta$) distribution, here, $\alpha = 1$, $\beta = 1$. In order to improve efficiency and increase the security of block-producing nodes, the nodes in the framework are divided into miner nodes and voting nodes. Voting nodes improve their trust degree by actively participating in the selection of miner nodes, which effectively avoiding the centralization of rights and interests. Miner nodes are selected by the way of trust competition, which can alleviate the waste of computing power caused by mining. Set the two parameters $\alpha$ and $\beta$ in beta distribution, the $\alpha$ and $\beta$ are dynamically updated in each round of consensus. Defining $\alpha$ as the number of honest signatures and $\beta$ as the number of malicious signatures, $\alpha > 0$, $\beta > 0$. The trust value of the node is defined as P, which consists of DP and IP. DP represents the initial trust, which is calculated from the previous historical communication of the node and IP represents the recommendation trust which is calculated from the comprehensive recommendation of all nodes in the neighborhood. Set the weight $\lambda$ to indicate the degree of trust in the initial and recommendation trust. We have the following Equation (1):

$$P_i = \lambda \times DP_i + (1 - \lambda) \times IP_i. \tag{1}$$

The initial trust of the node obeys beta distribution, and it still obeys beta distribution after dynamic parameter update. Using gamma distribution for beta distribution f (p | $\alpha$, $\beta$) is define as Equation (2):

$$f(p|\alpha, \beta) = \frac{\Gamma(\alpha+\beta)}{\Gamma(\alpha)\Gamma(\beta)} P^{\alpha-1}(1 - P)^{\beta-1}. \tag{2}$$

The initial trust value of the node is represented by the expected value of the probability distribution, so the expression is defined as Equation (3):

$$E(P) = \frac{\alpha}{\alpha+\beta}, \tag{3}$$

after each round of voting, the number of exchanges will be increased, and the node trust value will be dynamically updated. Broadcasting the updated trust status of the node to the entire blockchain as the initial trust value for the next round.

Voting produces two results: honest behavior and malicious behavior, each round of voting is the performance of nodes actively participating in block generation. Honest behavior can ensure that the selected nodes are more secure, and malicious behavior means that there are wrong nodes to undermine the overall security. After each round of voting, the connection between the nodes is re-established. Assuming that the number of honest votes cast by voting nodes are s and the number of malicious votes cast by voting nodes are f, the trust parameters of the previous node stored in the blockchain are updated, and the trust value of the node still obeys the beta distribution after updating. Therefore, the parameters $\alpha = s + 1$, $\beta = f + 1$, $DP_i$ will be updated as Equation (4):

$$DP_i = E(Beta(s + 1, f + 1)) = \frac{s+1}{s+f+1} \tag{4}$$

The trust value parameters $(\alpha, \beta)$ updated after each round of consensus process are stored in the blockchain as the initial value of the next round. After N rounds of consensus, the $DP_i$ value of the target node is defined as Equation (5):

$$DP_i = \sum_{x=1}^{n} \frac{\eta \alpha_{x-1} + 1}{\eta \alpha_{x-1} + \theta \beta_{x-1} + 2}, \tag{5}$$

in Equation (5), x represents the number of times a node participates in consensus process. Taking into account the dynamics of node reputation, the time forgetting factors $\eta$ and $\theta$ are introduced. The higher the node parameter, the faster the accumulation of trust is prone to centralization problems. Therefore, the influence of historical reputation on recent reputation is adjusted by parameters. In order to make the entire system more stable, setting parameter $\eta \in (0.6, 0.8)$, which means encouraging honest voting nodes and setting parameter $\theta \in (0.2, 0.4)$, which means punishing malicious voting nodes and warning them to vote reasonably in the next round.

The direct trust value can be used to initially understand the behavior of nodes participating in block generation, but this one-stage trust value calculation cannot prevent malicious nodes from colluding and doing evil to obtain verification rights. Malicious nodes can still pass misinformation to the blockchain through collusion attacks, which reduces the credit value of some honest nodes, while the behavior of malicious nodes will not be discovered. Therefore, the node neighborhood is established, and the initial information is verified through the information obtained by other nodes in the neighborhood evaluating the target node. According to the characteristic that most nodes in the neighborhood do not have too much deviation in the evaluation of the trust value of the target node, once the node is found to be malicious, broadcasting this behavior in the next round of the process can well identify the malicious node and realize the rapid elimination of the recommendation list. At the same time, the node's equity can be dispersed through the recommendation of nodes in the neighborhood to avoid the occurrence of monopoly due to excessive credit of the node. High-credit nodes are attacked, which reduces the security of the overall blockchain. Therefore, the prior information obtained should be corrected according to the recommended information. It is computed as the following Equation (6):

$$IP_i = \frac{1}{m} \sum_{j=1}^{n} DP_{ij}, \tag{6}$$

where IP is the trust value obtained from the node recommendation list. The neighborhood is established according to the communication radius of the target node, and the initial trust value is verified by the points in the neighborhood during each round of consensus on the information recommended by the target node.

According to the dynamic update of the initial trust value in the first stage and the correction of the recommended information in the second stage, the comprehensive trust value of the target node is finally defined as Equation (7):

$$P_i = \lambda \sum_{x=1}^{n} \frac{\eta \alpha_{x-1} + 1}{\eta \alpha_{x-1} + \theta \beta_{x-1} + 2} + (1 - \lambda) \frac{1}{m} \sum_{j=1}^{n} DP_{ij}, \tag{7}$$

different $\lambda$ values can be set up according to different scenarios, so as to observe the changes in the growth of the node trust value curve. After obtaining the comprehensive trust value of the target node and broadcasting to the blockchain, each node will get the new credit status of the target node. Through the way of credit value competition on the chain, it can ensure that the selected nodes better maintain the overall security and stability of the blockchain.

### 3.2. PoT Consensus Mechanism

The public chain assumes that all nodes are untrustworthy, and the strategy for generating blocks is mainly based on the PoW mechanism of computing power. In the PoW mechanism, nodes compete with each other for mining. In order to obtain block rewards, a large amount of computing power must be paid, which will lead to block forks and cause overall instability. At present, the Bitcoin network generates a block every 10 minutes to obtain rewards. However, when the revenue is not enough to maintain the electricity cost of mining, miners no longer have enough motivation to maintain the consistency of the blockchain. Therefore, a PoT consensus mechanism is established, in which it rewards and punish the behavior of nodes in the voting process and combine the updated node trust value to compete on the chain instead of traditional mining.

### 3.3. Rewards and Punishment Mechanism

In this section, a reward and punishment mechanism is designed for node trust value assessment. Normal behavior of node, that is, compliance with the system rules, the credit value will gradually increase as the lamination of the consensus process. Conversely, the node of behavioral abnormalities will be immediately identified, broadcast to the entire block chain, and the credit value decreases with the consensus process. Before the reward and punishment mechanism are introduced in detail, we will first describe the two node behaviors in the block production process.

Good behavior: the signature of the node represents voting the node. In the voting process, the node uses its own private key to sign, and other nodes can verify with the public key. As long as the blocks generated by the node meet the system rules, they are eligible to compete in the chain, this behavior is defined as a good behavior, the node is honest node.

Bad behavior: there are two types of bad behaviors, namely node collusion and malicious voting. The former increases the activity in the blockchain by verifying a large number of transactions from a long time ago, in order to attracting a large number of nodes to vote. The latter promotes the credit value through the behavior of nodes colluding and slandering honest nodes. Even though such behavior will be detected by asynchronous consensus mechanism, it slows down the efficiency of blockchain system. This behavior is defined as a bad behavior, the node is malicious node.

Thus, according to the behavior of node i, we divide $P_i$ into two components, which can be defined as Equation (8):

$$P_i = \begin{cases} \left(1 + \sum_{k=1}^{n} \frac{k}{\Delta T}\right) P_i w & i \in good; \\ \sum_{k=1}^{n} \frac{k}{\Delta T} P_i & i \in bad, \end{cases} \tag{8}$$

where k denotes the number of valid transactions of node i during the latest unit of time, ΔT denotes a unit of time. In order to avoid the node credit value growth is too fast, set parameter w. According to past experience, six blocks in the PoW consensus mechanism confirm that the transaction writes into the block chain, so w = 0.8 is set to avoid node centralization. That is to say, if the node is in an active state during the consensus process, actively vote, the system will adjust according to the node behavior, ensure that such nodes can submit transactions faster, reduce electricity consumption. In contrast, according to the equation (8), the credit value will be rapidly reduced when the node has malicious behavior and cannot be restored to the initial state in a long period of time. When malicious behavior occurs, the system will add this type of node to mining, causing malicious nodes to continue to attack.

### 3.4. Trust Competition on the Chain

A consensus mechanism PoT based on credit value calculation is proposed. Nodes with high credit value can quickly reach a consensus output block, while malicious nodes will be punished, reducing credit value and increasing the difficulty of block generation. In this mechanism, adjusting the difficulty of acquiring blocks according to the behavior of nodes can effectively reduce the waste of computing power and reasonably resist external attacks, ensuring the security and stability of the whole blockchain. We state that the difficulty of mining is inversely proportional to the credit value. The credit value is dynamically adjusted in the whole consensus process, and the credit value of each node is updated under the action of the reward and punishment mechanism. Then, the PoT consensus mechanism is presented in Figure 2.

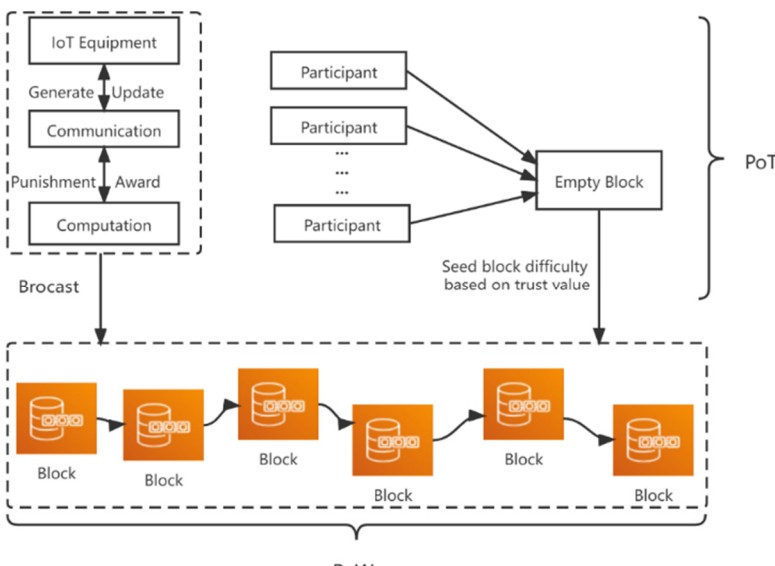

**Figure 2.** PoT consensus mechanism.

## 4. Results and Analysis

In this section, we have evaluated the performance of the credit-based PoT mechanism. Randomly inserting wrong nodes and malicious attacks are performed to verify the mechanism, in order to quickly eliminate malicious nodes through the recommendation of nodes and the behavior of nodes participating in block creation. We have evaluated credit-based PoT mechanism comparing to traditional PoW mechanism on performance. The article mainly simulates and verifies the following content:

(1) Set different initial trust value parameters to observe the growth of the target node's trust value.

(2) Increase the proportion of malicious nodes. Compare the difference between PoT and PoW in quickly removing malicious nodes.

(3) A comparative analysis of the cumulative growth of node trust value in the case of different proportions of malicious nodes.

(4) Comparative analysis of the consensus delay between trust value competition and traditional mining. The parameter settings and the obtained results will be presented and discussed below.

The results show that nodes in the PoT mechanism can quickly eliminate malicious nodes through credit value calculation, and the behavior of nodes actively participating in block construction can effectively reduce the consensus delay. Therefore, the solution is more widely applicable to the Internet of Things environment.

### 4.1. Trust Accumulation

Voting can accumulate the trust value of nodes, and the more honest nodes there are, the faster the trust value accumulates. Set the parameters of the target node $\alpha$ and $\beta$, and its trust value is 0.5 according to equation (3). According to our setting, the number of neighbors around the target node is 4, so after 20 rounds of consensus process, the number of votes accumulated by the node is 80, which includes honest voting and malicious voting. After a round of consensus process, the node parameters are updated to $\alpha = 71$, $\beta = 29$, and the node probability distribution is defined as Figure 3.

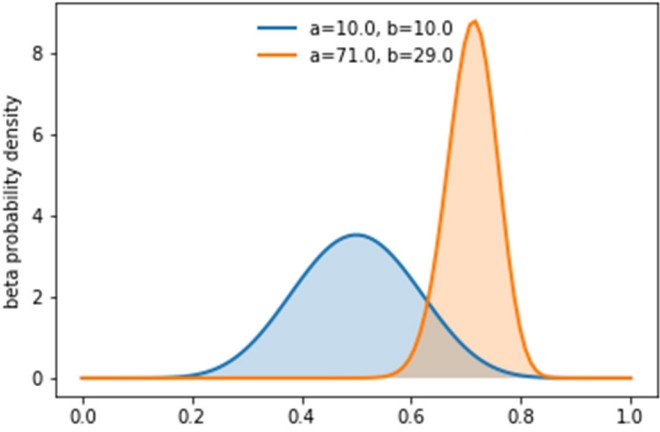

**Figure 3.** Beta probability distribution.

From the Figure 3, it can be seen that the probability distribution of nodes has moved from the initial 0.5 to 0.71. Therefore, through a large number of honest nodes voting, the trust value of nodes is improved, which is conducive to maintaining the overall security of blockchain.

### 4.2. Activity Influences

The more nodes interact with each other, the better they can understand each other's behavior during the block generation process, which is conducive to the improvement of trust value. The initial reputation of a node is based on historical communications stored in the blockchain. If a node is familiar to everyone before it becomes a block producer, it can attract more nodes in the voting process. In order to verify this point of view, the article did three sets of comparative experiments. The observability of one node is not obvious enough, so we pack the selected 20 nodes into an analysis of the overall trust value growth. The parameters are selected as $\alpha = 10$, $\beta = 10$, $\alpha = 30$, $\beta = 30$, $\alpha = 50$, $\beta = 50$ for analysis, the initial trust value of the nodes is all 0.5, and 20 nodes are packaged together, the initial trust value is 10. With the increase in the number of consensus rounds, it was found that the trust value of the first set of experiments in the environment lacking the number of interactions increased slowly and gradually became flat. The second and third groups of experiments showed a clear upward trend with the increase in the number of consensus rounds, and the cumulative trust value of this group of experiments with the parameters of $\alpha = 50$, $\beta = 50$ was higher than that of the parameters of $\alpha = 30$ and $\beta = 30$. It shows that

only through the accumulation of good reputation in the past can nodes attract more nodes to vote for them, so as to accumulate trust value and serve the entire community stably. Then, trust increase in network is presented in Figure 4.

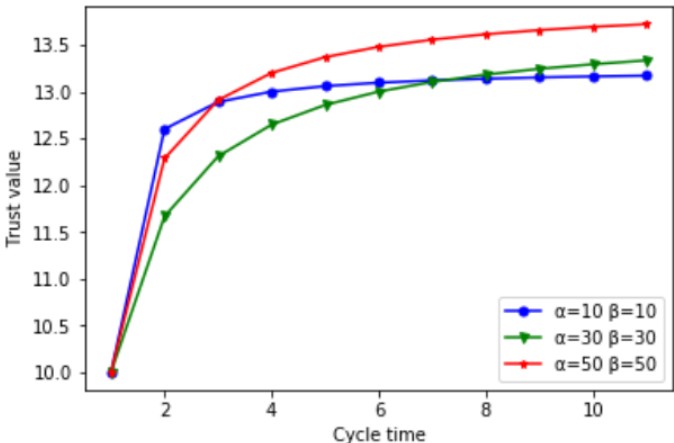

**Figure 4.** The effect of positivity on trust accumulation.

### 4.3. Malicious Identification

In this experiment, real-time attacks are carried out by increasing the proportion of malicious nodes, so as to analyze whether our proposed framework has the stability to quickly identify and remove malicious nodes. When a node is unknown, assume that the trust is 50%, so the initial trust value is 0.5. We randomly added the number of malicious nodes with a proportion of 30%, and observed whether the model could identify malicious nodes and provide security services as the number of consensus rounds increased. In order to observe the effectiveness of the proposed model, simulation was established and compared with the existing consensus mechanism PoW. Figure 5 describes the model removed all malicious nodes after the fifth round and reached a stable state.

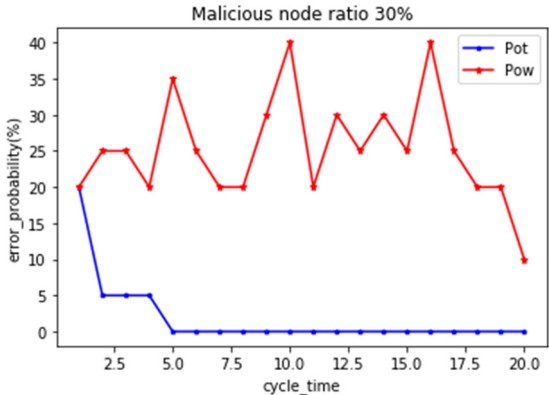

**Figure 5.** The malicious nodes account for 30%.

In order to verify the sensitivity of the model, we carried out multiple comparison experiments, and continued to compare the simulation experiments in which malicious nodes accounted for 40–60%. As shown in the Figure 6. When the proportion of malicious nodes increased to 40% and 50%, it was found that the mechanism proposed by us reached stability after the seventh round, which increased by two rounds of consensus process compared with the previous experiment. It shows that the increase of malicious nodes weakens the ability of honest nodes to a certain extent, and also does not exclude the possibility that some malicious nodes collude with each other in the previous rounds of consensus to show good behavior. However, after several rounds of consensus process, malicious nodes can still be removed to ensure the stability of the blockchain. In the

fourth group of experiments, when the number of malicious nodes reaches 60%, compared with the existing consensus mechanism PoW, our mechanism PoT can still maintain on-chain security.

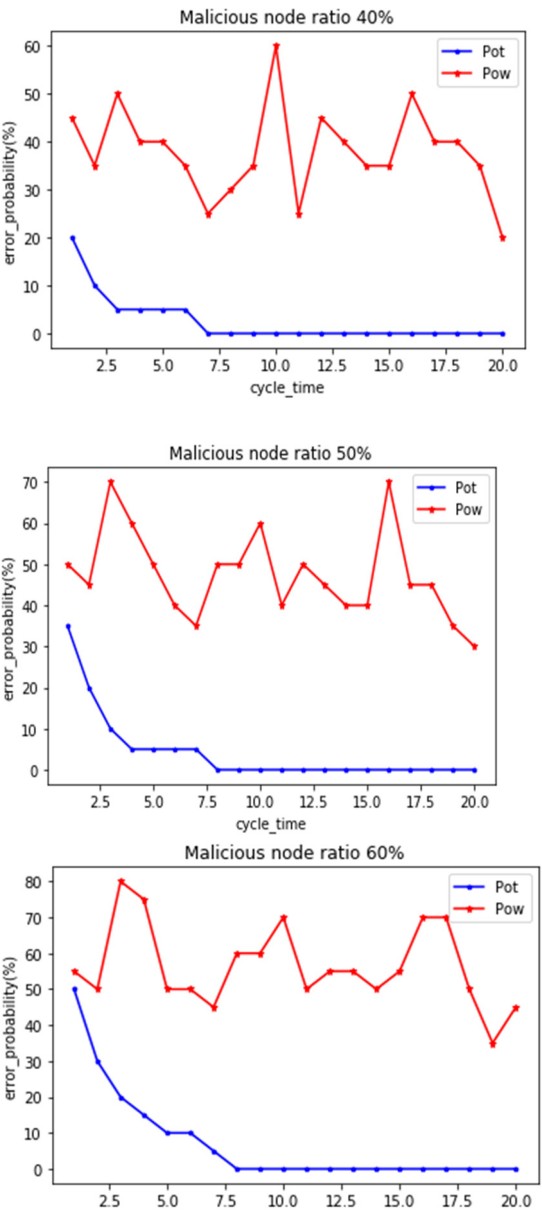

**Figure 6.** The malicious nodes account for 40–60%.

Summarizing, the model can effectively avoid nodes' malicious voting and real-time attacks by malicious nodes colluding and conspiring to obtain benefits. These attack nodes can be completely eliminated through several rounds of consensus mechanism, effectively maintaining the security of the chain and quickly reaching a stable state.

*4.4. Consensus Delay*

In this paper, a network with 20 block-producing nodes is built, and each node has 4 votes. The initial difficulty of mining is D = 6, and adaptive adjustment is made according to the behavior of nodes participating in the block. A total of 20 rounds of consensus are carried out Figure 7 describes the consensus delay variation of PoT consensus mechanism and PoW consensus mechanism. Consensus delay represents the block generation time in the blockchain network, and the delay unit is seconds (s). It is observed that the consensus

delay of PoW is always above PoT, indicating that the nodes in PoT obtain a large number of votes through honest behavior, compete for unchain based on trust, and adjust the difficulty of mining adaptively, which can effectively alleviate the waste and delay caused by traditional computing power mining.

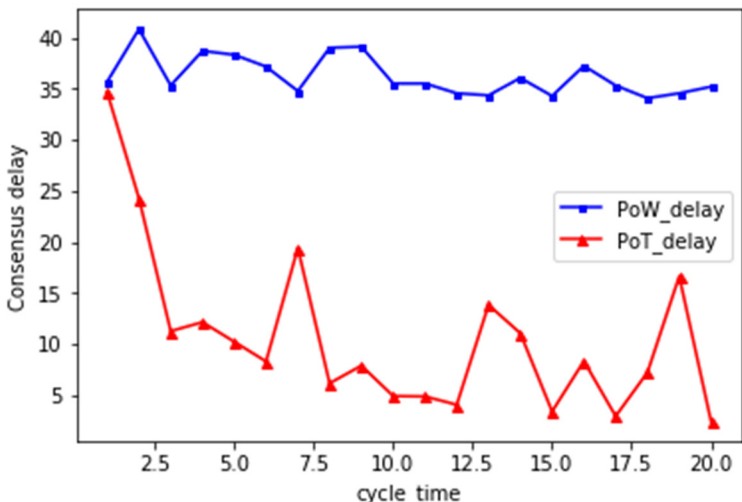

**Figure 7.** The consensus delay.

Based on the analysis of the above experimental results, the proposed framework can effectively identify the behavior of malicious nodes, and almost all malicious nodes will be detected and removed in the fifth round of consensus. In addition, the proposed mechanism can effectively reduce consensus delay and save computational power.

## 5. Discussion

The two-stage trust model can quickly identify and eliminate the wrong nodes, which can effectively solve the problem of node collusion in evil existing in the previous single-stage trust model. This paper proposes that nodes participating in the competition of block creation can effectively reduce the problem of consensus delay. A node's behavior instead of traditional mining can effectively alleviate the problem of computing power waste and consensus delay, providing new ideas for future research.

## 6. Conclusions

In this paper, an interaction scenario of IoT devices based on blockchain technology improvement is simulated. In this scenario, devices establish connections with each other to obtain valuable information. The interaction process involves resource consumption and value transfer. Therefore, interactive devices need to find trusted partners for communication and efficient information transfer to reduce delays.

To this end, we design a PoT-IoT framework. Under this framework, nodes participate in block creation behavior competition and go up the chain instead of traditional mining, which can effectively solve the problem of wasted computing power and consensus delay. In order to enable devices to transmit information securely in the process of IoT interaction, a two-stage trust computing model is introduced. Based on the local communication of nodes, historical information is obtained and broadcast to the blockchain. Neighborhood nodes observe their behaviors and recommend them to obtain a global trust value. The establishment of a reward and punishment mechanism can reward nodes that actively maintain blockchain security, and punish nodes that conspire to do evil and malicious attack nodes to prevent malicious nodes from gaining high trust values and attacking the blockchain in real time.

In order to verify the effectiveness of the proposed framework, a simulation experiment is carried out, and a comparative analysis is made with the existing consensus mechanism

PoW from the perspective of malicious node identification and consensus delay. However, the paper only simulates the IoT interaction experiment through simulation. Therefore, in future we will conduct research in real scenarios.

**Author Contributions:** Methodology, X.T. and F.S.; software, C.W.; validation, X.T. and C.Y.; formal analysis, F.G.; investigation, H.C.; resources, X.T.; data curation, X.T.; writing—original draft preparation, X.T.; writing—review and editing, F.S. and F.G.; visualization, C.W. and C.Y. All authors have read and agreed to the published version of the manuscript.

**Funding:** This research was funded by 1. [Beijing Social Science Fund] grant number [20GLB028]; 2. Beijing Municipal Universities Basic Research Funds, Capital University of Economics and Business [ZD202105].

**Institutional Review Board Statement:** Not applicable.

**Informed Consent Statement:** Not applicable.

**Data Availability Statement:** Not applicable.

**Conflicts of Interest:** The authors declare no conflict of interest.

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
