# Peer review of "Trusted Blockchain-Driven IoT Security Consensus Mechanism"

_sustainability, doi:10.3390/su14095200_

Round 1

Reviewer 1 Report

The manuscript is generally interesting and well written. The topic of the paper is connected with the recently popular issue of blockchain technology. The level of research is appropriate for publication.

The structure of the paper is also correct. The subsequent chapters connect with each other logically. The text is finished with conclusion of the analysis.

The references were also chosen correctly. I can see 32 references, all of them are quite new (only one was published before 2017).

However, I have two comments that should e taken into account by authors:

  • Problem of language. Generally, it is not so bad. But in many places authors use "we", "our". It is technical text, so in such documents they should use passive voice. No "we did", but "it was done". I think it should be corrected.
  • One element to check (or explain); in text you use once: α (alpha) and β (beta), then a and β (lines 207 and 360). How should it be? Please, check it.

Reviewer 2 Report

The paper titles “Trusted blockchain-driven IoT security consensus mechanism” is based on the observation that normally the information interacted by IoT devices is stored through third-party intermediaries, but the cost of centralized storage is high and there is a risk of data leakage. Also, authors highlighted that the single point of failure and node attack tend to cause instability in the centralized system which is a valid point.  The main work proposed a framework which is inspired by the problem of  existing blockchain consensus mechanism of wasting of computing power.

The framework based on a two-stage credit calculation to handle these problems. Notably, the nodes are selected through the model, and these nodes will compete on the chain according to the behavior of participating in the creation of the block.

Removal of  malicious nodes, maintain the overall security of the blockchain and reduce consensus delay is shown by comparative simulation which favor the proposed framework.

The work is interesting and addressed a problem of practical  interest.

In order to make the entire system more stable, setting parameter is used but it is now clear why  malicious voting nodes are still allowed with  warning them to vote reasonably in the next round? It is possible to demote them more aggressively.

Some related work can improve the work by considering the real-time nature of attacks. For refence, see: An application of markov jump process model for activity-based indoor mobility prediction in wireless networks

-Frontiers of Information Technology, A goal programming based energy efficient resource allocation in data centers-The Journal of Supercomputing ,Comparison and analysis of greedy energy-efficient scheduling algorithms for computational grids-Energy Aware Distributed Computing Systems, Utilization bound for periodic task set with composite deadline-Computers & Electrical Engineering, Lowest priority first based feasibility analysis of real-time systems-Journal of Parallel and Distributed Computing, Cost efficient resource allocation for real-time tasks in embedded systems-Sustainable Cities and Society, Diverse routing in multi-domain optical networks with correlated and probabilistic multi-failures-IEEE International Conference on Communications, Minimizing response time implication in DVS scheduling for low power embedded systems- Innovations in Information Technologies (IIT), A robust iris localization scheme for the iris recognition-Multimedia Tools and Applications.

Conclusion needs rewrite. There seems no difference between abstract and conclusion. It is suggested not to used statements like “there are still some limitations in our systems”, try to write concrete  terms. For instance, “some limitation” does not reflect what authors want to mention in the context.

It seems the authors have already discussed the impact of and seems the information is reductant in Figure 3. Beta probability distribution.
